# Reference Values for External and Internal Training Intensity Monitoring in Young Male Soccer Players: A Systematic Review

**DOI:** 10.3390/healthcare9111567

**Published:** 2021-11-17

**Authors:** Rafael Oliveira, João Paulo Brito, Adrián Moreno-Villanueva, Matilde Nalha, Markel Rico-González, Filipe Manuel Clemente

**Affiliations:** 1Sports Science School of Rio Maior, Polytechnic Institute of Santarém, 2140-413 Rio Maior, Portugal; jbrito@esdrm.ipsantarem.pt (J.P.B.); matildenalha@gmail.com (M.N.); 2Research Center in Sport Sciences, Health Sciences and Human Development, 5001-801 Vila Real, Portugal; 3Life Quality Research Centre, 2140-413 Rio Maior, Portugal; 4Department of Physical Activity and Sport Sciences, International Excellence Campus “Mare Nostrum”, Faculty of Sports Sciences, University of Murcia, 30720 San Javier, Spain; more_adri@hotmail.com; 5Department of Didactics of Musical, Plastic and Corporal Expression, University of the Basque Country, UPV-EHU, 48940 Leioa, Spain; markeluniv@gmail.com; 6Escola Superior Desporto e Lazer, Instituto Politécnico de Viana do Castelo, Rua Escola Industrial e Comercial de Nun’Álvares, 4900-347 Viana do Castelo, Portugal; filipe.clemente5@gmail.com; 7Instituto de Telecomunicações, Delegação da Covilhã, 1049-001 Lisboa, Portugal

**Keywords:** football, male, training, youth, RPE, GPS, running, high-speed running, sprint

## Abstract

Training intensity monitoring is a daily practice in soccer which allows soccer academies to assess the efficacy of its developmental interventions and management strategies. The current systematic review’s purpose is to: (1) identify and summarize studies that have examined external and internal training intensity monitoring, and to (2) provide references values for the main measures for young male soccer players. A systematic review of EBSCO, PubMed, Scielo, Scopus, SPORTDiscus, and Web of Science databases was performed according to the Preferred Reporting Items for Systematic Reviews and Meta-Analyses (PRISMA) guidelines. From the 2404 studies initially identified, 8 were fully reviewed, and their outcome measures were extracted and analyzed. From them, the following range intervals were found for training: rated perceived exertion (RPE) 2.3–6.3 au; session-RPE, 156–394 au; total distance, 3964.5–6500 m and; distance >18 km/h, 11.8–250 m. Additionally, a general tendency to decrease the intensity in the day before the match was Found. This study allowed to provide reference values of professional young male players for the main internal and external measures. All together, they can be used by coaches, their staff, or practitioners in order to better adjust training intensity.

## 1. Introduction

Training intensity monitoring is a daily practice within the soccer training context [1]. Training intensity describes how hard a player is exercising [2]. Understanding the impact of training stimulus on the players and identify the variations between players contributes to an adjustment of planning and for adjusting specific recovery or managing strategies [3]. For that reason, coaches expect that training intensity monitoring guides them for planning better for an improvement in performance and reduction in injury risk [1]. Quantifying aspects of young players development and performance (e.g., physical abilities, physiological capacities, technical and tactical skill) allows soccer academies to assess the efficacy of its developmental interventions and management strategies [4]. By collecting and analyzing regular inputs related to player development and performance, organizations create a feedback loop in which their planning and training interventions can be objectively assessed. 

The monitoring is commonly organized in two dimensions [5]: (i) internal intensity; and (ii) external intensity. The internal intensity represents the psychophysiological impact of external intensity on the player’s organism while the external intensity represents the mechanical and physical impact of training drills on the players [6]. Different instruments can be used to assess these dimensions, however in soccer, the most common instruments for measuring internal intensity are heart rate monitors and rated perceived exertion (RPE) scale, while for measuring the external intensity the most common are microelectromechanical systems (e.g., global positioning system, local positioning system, inertial measurement units) [7]. 

From such instruments, the typical variables extracted are heart rate-bases scores, RPE-based scores and distances covered at different speed thresholds, or changes of velocity (accelerations and decelerations). The measures are daily quantified in training and match scenarios. Using absolute intensity per session and per week, the sports scientist has some possibilities of understanding the within- and between-weeks variations [7]. 

The analysis of intensity impact on players is highly relevant. One of the reasons is justified by the gap between coaches perception and the actual intensity imposed on the players [8]. Based on that, employing intensity monitoring in youth teams became a typical strategy used by coaches [9]. Thus, a growing number of researches have been published in training monitoring in youth soccer players [10,11,12]. 

Although the increase in intensity monitoring reports in young soccer, most of the articles are from the same team. This reduces the possibility of generalization of the evidence. However, is also known that is not expectable to reach different teams at the same time over long periods of the season. Thus, one of the strategies can be to summarize the available evidence about intensity values of young soccer players by conducting a systematic review. Although a possible idea, no systematic review was found about that. Although there is pertinence of looking into both males and females, summarizing both within the same article would not be useful since the huge differences related with a number of publications and a logical difference between sexes produces a different approach to the consequences for the article logic. Thus, a more focused contribution as a systematic review may help coaches to compare values of training intensity in young male soccer and provide some benchmarks. Thus, the aim of this systematic review is to identify and summarize studies that have examined external and internal training monitoring and to provide references values for the main measures for young male soccer players.

## 2. Materials and Methods

The preferred reporting items for systematic reviews and meta-analyses (PRISMA) guidelines were followed to write this systematic review [13] and guidelines for performing systematic reviews in sport sciences [14]. The protocol of the systematic review was a priori registered in the International Platform of Registered Systematic Review and Meta-Analysis Protocols with the number INPLASY202180055 and the DOI number 10.37766/inplasy2021.8.0055.

### 2.1. Eligibility Criteria

The inclusion and exclusion criteria can be found in Table 1.

The screening process related to analysis of the title, abstract and reference list of each study to locate potentially relevant studies was independently executed by two of the authors (A.M.V. and M.R.G.). Moreover, both authors also reviewed the full version of the included papers in detail to identify which article met the inclusion criteria. Additionally, a search within the reference lists of the included records was performed to add additional relevant studies. In the cases of discrepancies, a discussion was performed with the participation of a third author (R.O.). Possible errata for the included articles were considered.

### 2.2. Information Sources

The following electronic databases were used to search for relevant publication on the 3 July 2021, after protocol registration: FECYT (MEDLINE, Scielo, and Web of Science), PubMed, and Scopus. A manual search was also conducted after search in electronic databases to retrieve additional studies that could fit our eligibility criteria. 

### 2.3. Search Strategy

Keywords and synonyms were entered in various combinations in the title, abstract or keywords which means that the following research content was applied: (“soccer” OR “football”) AND (“internal load” OR “external load” OR “workload” OR “training load” OR “load monitoring”). Search results were exported to EndNote 20.0.1 for Mac (Clarivate Analytics). No filters or limits were applied.

### 2.4. Data Extraction

A specific spreadsheet was designed in Microsoft Excel (Microsoft Corporation, Readmon, WA, USA) for process the data extraction. The design followed the recommendations of the Cochrane Consumers and Communication Review Group’s data extraction template [15]. In this spreadsheet the information about inclusion and exclusion requirements and reasons was detailed. The selection of the articles was made independently by two authors (A.M.-V. and M.R.-G.). In the cases of discrepancies, a discussion was performed with the participation of a third author (R.O.).

### 2.5. Methodological Assessment

The methodological quality was assessed using the methodological index for non-randomized studies (MINORS) by two authors (A.M.-V. and M.R.-G.) [16]. MINORS consists of twelve items, four of which are only applicable to comparative studies which was not the case of the included studies. Thus, only eight items were applied. Each item is rated as 0 when the criterion is not reported in the article, 1 if reported but not sufficiently fulfilled, or 2 when adequately met. Higher scores indicate good methodological quality of the article and a low risk of bias. The highest possible score is 16 for non-comparative studies. MINORS has yielded acceptable inter- and intra-rater reliability, internal consistency, content validity, and discriminative validity [16,17]. 

## 3. Results

A total of 2404 (i.e., FECYT: 1481; PubMed: 806; Scopus: 117) original articles were initially retrieved from the mentioned databases, of which 834 were duplicates. Thus, a total of 1570 original articles were found. After this, a total of 1558 articles checked by title and abstract were excluded. The remaining 12 articles were checked in full text, leading to the exclusion of 3 articles according to criterion nº 1, and 1 according to criterion nº 2. Additionally, 1 article was included from additional sources. A total of 8 articles met all the inclusion criteria and were finally included in the qualitative synthesis. All the steps followed for the selection of the articles are available in Figure 1.

### 3.1. Methodological Quality

The overall methodological quality of the cross-sectional studies can be found in Table 2.

### 3.2. Results of the Studies

Table 3 presents the characteristics of the studies regarding their sample size, their age of the sample, their duration, as well as training duration, internal and external measures, and instruments used. From the eight studies included, the study of Hannon et al. was the only to analyze the categories of under12 (U12), U13 [21]. Two studies analyzed U14 category [21,25]. Four studies analyzed U15 category [18,20,21,24]. Two studies analyzed the category of U16 [21,25]. Three studies analyzed the category of U17 [18,23,24]. Two studies analyzed the category of U18 [21,25]. Finally, three studies analyzed the category of U19 [18,22,24]. 

In addition, three studies analyzed both internal and external measures [18,19,24], three studies only analyzed internal measures [20,23,25] and two studies only analyzed external measures [21,22].

Table 4 presents the results for external measures in which four studies were included [18,19,22,24]. The table was organized according to the age categories.

Only one study presented data for starters and non-starters. Since non-starter presented lower values than starters, we provided a range interval with lower values related to non-starters and higher values related to starters [19]. 

The last line of Table 4 presents the range intervals for the external measures most used.

Table 5 presents the results for external intensity by match-day minus (MD-) and by players positions (last two lines). The approach of MD- was used by two studies [21,24], while player positions was analyzed by one study [24]. The last line of Table 5, before player positions data, presents the range intervals according to the MD- approach for the measures most used.

Table 6 presents results for internal intensity in which one study included 1-match week and 2 matches-week analysis [20], one study analyzed player positions [23], one study analyzed player status (starters vs. non-starters) [19], three studies analyzed the average overall team [24,25,26] and two studies used MD- approach [23,24]. The session-rated of perceived exertion (s-RPE) was the variable most used to quantify internal training intensity. Finally, one study presented internal intensity for gym training [25]. The main results showed a range interval range of 2.3 to 6.3 arbitrary units (au) for RPE between U14, U15, U16, and U18 [20,25] and 156–394 au for s-RPE between U15 and U17 [20,23]. 

## 4. Discussion

The aim of this systematic review was to identify and summarize studies that have examined external and internal training intensity monitoring in young male soccer players and to provide references values for the main training intensity measures. The main results showed the following range intervals by overall teams that include (U12 to U19):
training duration of 79–117 min [18,19,21,22,24];total distance of 364.5–6500 m [18,19,21,22,24];distance > 18 km/h of 11.8–250 m [18,19,21,22];distance > 25 km/h of 0–30 m [18,19,21,22,24].

The importance of prescribing appropriate training intensities to improve player performance is well recognized in the current literature [27,28]. However, there remains a lack of clarity regarding the training intensity values most likely to promote improvements in young players performance. 

### 4.1. Reference Values Depending on Age Group

When analyzing the mean of the total distance per training session for overall team of the selected studies, there was a pattern of increasing distance until older ages [18,21,22,24] which was also corroborated by increase in the number of body impacts [18]. However, Dalen and Lorås reported a decrease in the number of accelerations from U15 to U17, respectively, [19], which was in line with Teixeira et al. that found the same pattern from U17 to U19 [24].

When considering, the accumulated weekly, total distance did not show that pattern from under U12 (~18.6 Km), U13 (~19.3 km), U14 (~19.5 km), U15 (~21.0 km), U16 (~20.8 km), U17 (~16.0 km), U18 (~21.2 km) to U19 (~16.0 km) age-groups [18,19,21,22,24,25]. Considering only the study of Hannon et al. that quantifying the training and match volume in male players from an English Premier League academy during two in-season microcycles, it was found an increase in the accumulative total distance from U12/13 (38.3 ± 5.1 km), to U15 (53.7 ± 4.5 km) and to U18 (54.4 ± 7.1 km) [21]. Additionally, Abade et al. reported a higher total distance covered for U17 players than for U19 [18]. 

Perhaps technical-tactical methodologies, namely game model, can explain these data. There was an age-related increase in the training intensity and to a greater extent in the training volume [25]. Due to this fact, associated with a more conscious pacing strategy and better game interpretation with age, it was possible to also increase the exercise economics [24,29]. Even more interaction effects were found between inter-day and age [21,24], confirming an increase in the pacing strategy in the aging progression. Training periodization also seemed to influence the external intensity, concerning the training day and weekly microcycle [24].

The sprints number reported by the studies showed a high dispersion. In the study by Abade et al. [18] 11 ± 6 sprints (distance = 12 ± 5 m) were reported in U15 but in Teixeira et al. [24] only 2 ± 3 sprints (sprint distance = 28 ± 42 m), however the distance is twice that reported by Abade et al. [18]. In U17, the sprint number and sprint distance data were, respectively, between 16.4 ± 8.2 (sprint distance = 13.0 ± 5.3 m) [18] and 4.8 ± 4.8 (sprint distance = 130.4 ± 462.6 m) [24]. Teixeira et al. (2021) study evidenced the lowest intensity in U15 players’ training sessions regarding high-speed run, average sprint distance, number of sprints [24]. The high standard deviation expresses the dispersion of results, which makes it difficult to standardize training intensity patterns related to the sprint number and distance at youth age.

Regarding internal training intensity, the main results showed a range interval range of 2.3 to 6.3 au for RPE between U14, U15, U16, and U18 [20,25] and 156 to 394 au for s-RPE between U15 and U17 [20,23]. Additionally, after conducting the research analysis of the present systematic review, a new study in U17 soccer players that analyze RPE and s-RPE measures was published [30]. That study [30] is in line with the interval range for RPE but higher values of 640 and 595 au were found for s-RPE during pre- and in-season with training durations around 96 and 95 min, respectively, which may justify the higher values. 

Moreover, Wrigley et al. [25] noted a higher weekly RPE in the older age group (i.e., U18). However, the authors Teixeira et al. verified a higher training volume in younger players (i.e., U15 vs. U19) [24]. It is reasonable to argue that coaching team tends to code training programs with more volume and less intensity when it comes to younger players [24,31,32]. Furthermore, a focus on the basic tactical principles and technical skills using constrained training tasks was reported in younger age groups [25]. Nevertheless, the time spent at high-intensity zones and normalizing the session duration may affect the perceived exertion [33].

There were only one study analyzed Banister TRIMP and player status and found no differences between starters and non-starters [19] which was also corroborated by Martins et al. [30]. Nonetheless, it is important to reinforce that the period of the season and microcycle can influence result interpretations. For instance, the comparison of RPE between starters and non-starters showed significant differences in the following day after the match due to the recovery session for starters. Additionally, some differences were found during some mesocycles of the in-season [30].

### 4.2. Training Intensity by Match Day Minus

The main findings showed that young players usually training between 3 to 4 days per week. The higher intensity was found on MD-4, MD-3, and MD-2 while the lower intensity was found on MD-1, although only two studies used this approach for data analysis [21,24]. The previous findings were similar to adult players [34,35,36,37,38,39] which seems to be convergent in a tapering strategy based on a gradual reduction until the last day before the match [40].

The decrease in high-speed running distance and sprint distance in training sessions before match day was also evident [21] but it was not confirmed in sprint distance by Teixeira et al. [24]. Some studies reported that some coaches use sprinting and acceleration exercises for neuromuscular activation as a pre-match activation methodology [24,40].

Regarding internal intensity analysis, two studies analyzed s-RPE with different scales, CR-10 [23] vs. CR-20 [24] and different training schedules, 4 [23] vs. 3 training sessions [24]. Even so, Nobari et al. found lower intensity in the following day of the match, higher intensity in mid-week, and the lowest intensities in the day before the match [23]. Although the study of Martins et al. [30] was in line with these findings, no differences between training days were found in Teixeira et al., study [24]. By contrast, previous studies reported an decreasing intensity phase in young players concerning RPE values [25,41] and Wrigley et al. (2012) evidenced a tapering in U18 players [25].

### 4.3. Player Positions

In youth soccer, the positional role has been analyzed in constrained training tasks [32,42,43], however, the present systematic review only found one study that analyzed intensity by position [24]. The study by Teixeira et al. analyzed three age-groups studied (U15, U17, U19) and reported that the greatest total distance was performed by the central midfielder (5456.9 ± 1565.9 m), followed by the wide midfielder (5370.1 ± 1692.6 m) [24]. The same authors only found significant differences between central defenders vs. forwards in high-speed running and sprint distance (minimum to moderate effect). Additionally, the internal training intensity presented significant differences between wide midfielder and forwards players (minimum effect). The same authors also reported an interaction effect between age, week, training day and playing position for deceleration [24].

The influence of playing position on physical and physiological performance during competition is well documented [44,45,46], while in training it seems to have a minimal effect for young players. In contrast, the influence of playing position in the adult training football has been well documented but it was non-conclusive [36,47,48,49]. On one hand, it was found greater training intensity for wide defenders and wide midfielders with respect to high-speed running and number of sprints when compared with the other positions [48]. In the same line, it was found in several mesocycles during an in-season, that central midfielders and wide midfielders displayed higher training intensity than central defenders with respect to total distance and high-speed running (>19 km.h^−1^) [49]. Other study found that midfielders had the highest training weekly acute load of high metabolic load distance (6901 AU), while central defenders had the lowest (4986 AU) [47]. On the other hand, it was found no differences between players positions for total distance, high-speed running (>19 km.h^−1^) [36] in mesocycle and microcycle analysis during an in-season. The results of the present review seems unclear regarding the influence of player positions on training intensity. A possible explanation could be related to the training tasks, which may not be representative of the positional role specificity as referred by Ferraz et al. [29]. In future studies, the weekly training intensity quantification should consider the game model and representative game-based situations to promote playing position specificity. Furthermore, speed and acceleration thresholds in the studies of this review were based on elite gold-standard guidelines. Future research should focus on the adjustment thresholds for elite and sub-elite youth football.

In our literature review, only two studies reported values of the internal training intensity between playing positions [23,24]. Teixeira et al. Reported significant differences [24], however Nobari et al. found no differences in the comparisons within weekdays between the playing positions [23]. In addition, and Gjaka et al. also found no differences in accumulated weekly intensity between player positions [20]. If data from adult training were considered, the results seem to be in line because, it was found in several mesocycles during an in-season, that central defenders displayed higher training intensity than strikers or central midfielders with respect to s-RPE [49], while higher values of s-RPE were reported for midfielders than other positions [50]. On the other hand, no differences were between players positions for the same variable [36] in mesocycle and microcycle analysis during an in-season. This information revealed that a non-consensus still remain which suggest future studies to confirm the results.

### 4.4. Study Limitations and Future Directions

This study presents some limitations that influenced the results and the reference values provided. First, the small number of studies proves that much more research is needed in young players, even more considering the different age-group categories. Additionally, the studies included only analyze one soccer team which limit their results. Consequently, few studies analyzed contextual variables, such as player status, player positions, and MD- approach. In addition, few studies included full seasons. Moreover, the different speed thresholds and scales used makes difficult to generalize the results and compare them between studies. In addition, training intensity and reference values provided came from general training, but this information was not revealed by the studies. Only one study identify data from the field and from the gym training [25]. Finally, the present systematic review did not consider some possible inter-country variability or the context of each study which should be considered in future studies.

Despite the limitations, the present study constitutes a relevant tool in the field of training intensity quantification of young male soccer players that can be used for coaches, their staff and practitioners as a reference for future studies in order to replicate such values or even to increase the numbers presented.

In future studies, the comparison between elite and sub-elite soccer academies is an important research gap which should be considered. Additionally, information on what type of training and exercises could provide further knowledge on training intensity. At last, future studies can analyze values of training intensity for injury prevention, since the general aims of the studies in this field are to improve performance and avoid injuries.

## 5. Conclusions

The standard microcycle from U12 to U16 included three training sessions, while U17 and U19 included four training sessions per week. The duration of training sessions for all age-groups was between ~70 and 117 min.

Specifically, the following range intervals were found for training: RPE 2.3–6.3 au; s-RPE, 156–394 au; total distance, 3964.5–6500 m and; distance > 18 km/h, 11.8–250 m.

Even though the internal training intensity did not present significant differences for weekly inter-day analysis, there was an inverted U-shaped curve in the distribution of the weekly external training intensity. In the older age-group players, tapering is evidenced. On the other hand, the U14 to U16 players seems present relatively high training intensities across the weekly microcycle. Indeed, it could be suggested that coaches opt for different tapering strategies when the age and competition focus increases.

The playing position seems to have a minimal effect on the weekly training intensity of young players.

## Figures and Tables

**Figure 1 healthcare-09-01567-f001:**
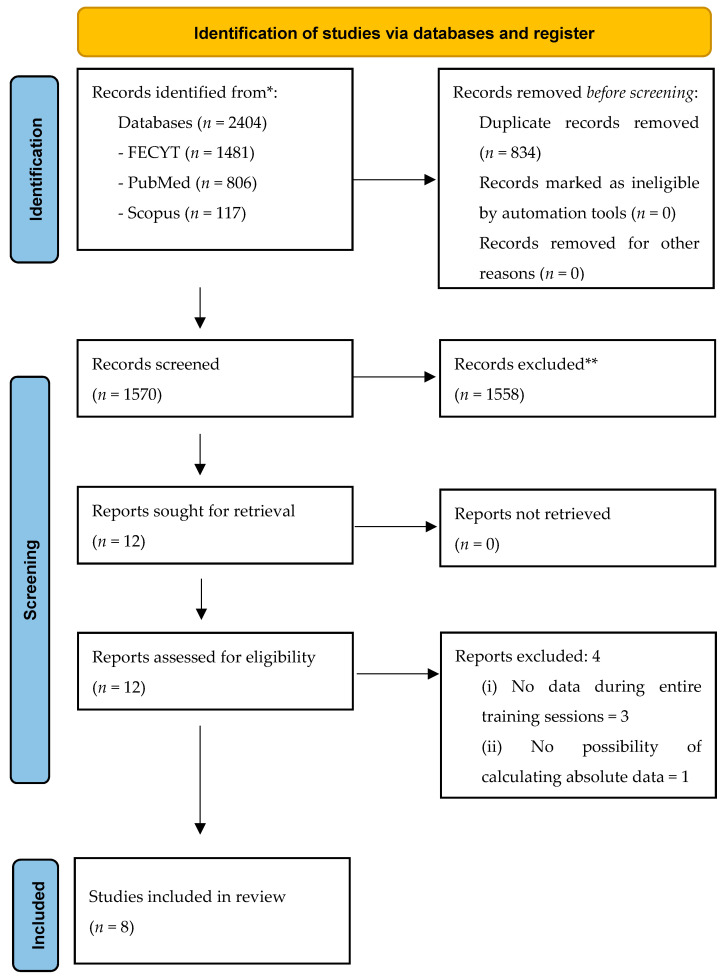
Preferred reporting item for systematic reviews (PRISMA) flow diagram.

**Table 1 healthcare-09-01567-t001:** Eligibility criteria.

PICOS	Inclusion Criteria	Exclusion Criteria
Population	Healthy young (under eighteen) male soccer players.	Studies conducted with professional or amateur players or in other sports, or with female populations.
Intervention/Exposure	Exposure to entire training sessions for number of weeks and sessions included (minimum one week).	No exposure to entire training sessions (e.g., specific exercises only reported; only matches; only simulated matches).
Comparator	Not required. Eventually, comparisons between playing positions and/or competitive levels within the same age-group and/or age-groups.	No study will be excluded based on comparators.
Outcomes	Presents at least of one measure among the included in internal (e.g., heart rate, rate of perceived exertion) and/or external intensity (e.g., distances covered at different speed thresholds, acceleration-based measures) in absolute values. Whenever relative values allow to calculate absolute values, the study will be included.	Absence of data characterizing the intensity during the training sessions (e.g., wellness variables, readiness parameters) and or only reports the data in relative values without allowing the calculation of absolute values. Data from workload calculations will also be excluded (e.g., accumulated weekly intensity, training monotony, strain, ACWR, EWMA).
Study design	No restrictions imposed on study design.	No study was excluded on the basis of study design.
Others	Only original and full-text studies written in English.	Written in other language than English. Other article types than original (e.g., reviews, letters to editors, trial registrations, proposals for protocols, editorials, book chapters and conference abstracts).

PICOS: (P) population; (I) intervention/exposure; (C) comparator; (O) outcomes; (S) study design.

**Table 2 healthcare-09-01567-t002:** Methodological assessment using MINORS checklist.

Study	1	2	3	4	5	6	7	8	Score
[18]	2	2	1	1	1	2	2	1	12/16
[19]	2	2	2	2	0	1	2	1	12/16
[20]	2	2	1	1	1	2	2	2	13/16
[21]	2	2	2	2	1	2	2	2	15/16
[22]	2	2	2	2	1	1	2	2	14/16
[23]	2	2	1	2	0	2	2	1	12/16
[24]	2	2	2	2	1	2	2	2	15/16
[25]	2	2	2	1	0	1	2	1	11/16

Note: The MINORS checklist asks the following information (2 = High quality; 1 = Medium quality; 0 = Low quality): 1. Clearly defined objective. 2. Inclusion of patients consecutively. 3. Information collected retrospectively. 4. Assessments adjusted to objective. 5. Evaluations carried out in a neutral way. 6. Follow-up phase consistent with the objective. 7. Dropout rate during follow-up less than 5%. 8. Appropriate statistical analysis.

**Table 3 healthcare-09-01567-t003:** Study characteristics.

Study	Sample	Age	Study and Training Duration	Internal Measures/ Instruments	External Measures/ Instruments
[18]	U15 N:56 U17 N:66 U19 N:29	14.0 ± 0.2 15.8 ± 0.4 17.8 ± 0.6	9 weeks Training duration: 90 min	1 Hz Polar Team System, Polar, FI: HR < 75%; HR 75–84.9%; HR 85% to 89.9%; HR ≥ 90%	15 Hz GPS (SPI-Pro X II, GPSports, Canberra, Australia): Total distance Distance 0–6.9 km/h Distance 7.0–9.9 km/h Distance 10.0–12.9 km/h Distance 13–15.9 km/h Distance 16–17.9 km/h Distance ≥ 18.0 km/h Distance ≥ 18.0 km/h NR Impacts NR
[19]	U17 N:18	15.7 ± 0.5	38 training sessions Training duration ND	1 HZ Polar Team Pro tracking system: HR	10 Hz Polar Team Pro tracking System: Total distance, Distance 11–14.9 km/h Distance 15.0–18.9 km/h Distance ≥ 19.00 km/h Acceleration ≥ 2.0 m.s^−2^ NR
[20]	U15 N:22	14.5 ± 0.3	4 weeks Training duration ND	s-RPE (CR-10)	-
[21]	U12 N:15 U13 N:13 U14 N:12 U15 N:10 U16 N:11 U18 N:15	11.7 ± 0.2 12.6 ± 0.3 13.7 ± 0.2 14.5 ± 0.3 15.5 ± 0.2 17.0 ± 0.4	1 in-season Training duration: described in Table 5	-	10 Hz GPS (Apex, STATSports, Northern Ireland): Total distance Distance 19.8–25.2 km/h Distance > 25 km/h
[22]	U19 N:9	17.6 ± 0.6	1 in-season Training duration 79 min	-	10 Hz GPS OptimEye X4 (Catapult Sports, Melbourne, Australia): Total distance Distance 12–15 km/h Distance 15–20 km/h Distance 20–25 km/h Distance > 25 km/h
[23]	U17 N: 21	16.1 ± 0.2	36 weeks Training duration ND	s-RPE (CR-10)	-
[24]	U15 N:20 U17 N:20 U19 N:20	13.2 ± 0.5 15.4 ± 0.5 17.39 ± 0.55	2 Weeks Training duration 90 min	RPE (CR-20) s-RPE (CR-20)	18 Hz GPS STATSports Apex^®^ (Newry, Northern Ireland): Total distance Maximal speed HMLD (>25.5 W·kg^−1^) Distance > 25 km/h Distance > 25 km/h NR Accelerations ≥ 3 m.s^−2^ Decelerations ≤ −3 m.s^−2^
[25]	U14 N:8 U16 N:8 U18 N:8	13 ± 1 15 ± 1 17 ± 1	2 Weeks Field training duration: U14—90 min U16—102 min U18—104 min Gym training duration: 30–35 min	5 Hz Polar Team System^®^, Kempele, Finland): HR s-RPE (CR10)	-

U: under; min: minutes; HR: heart rate; NR: number; ND: non described; HMLD: high metabolic load distance; RPE: rated perceived exertion; s-RPE: session rated of perceived exertion.

**Table 4 healthcare-09-01567-t004:** Results for external training by overall team.

Study	Measures	U12	U13	U14	U15	U16	U17	U18	U19
[18]	Total distance (m)	-	-	-	3964.5 ± 725.4	-	4648.3 ± 831.9	-	4212.5 ± 935.4
	Distance ≥ 18.0 km/h (NR)				10.9 ± 6.3		16.4 ± 8.2		11.8 ± 7.9
	Distance ≥ 18.0 km/h (m)				12.1 ± 4.9		13.0 ± 5.3		11.8 ± 6.7
	Impacts (NR)				490.8 ± 309.5		584.0 ± 363.5		613.1 ± 329.4
[19]	Distance 15.0–18.9 km/h (m)	-	-	-	-	-	~380–400	-	-
	Distance ≥ 19.00 km/h (m)						~180–200		
	Acceleration ≥ 2.0 m.s^−2^ (NR)						~65–75		
[21]	Duration (min)	~83–90	~85–100	~83–100	~85–110	~87–117	-	~70–115	-
	Total Distance (m)	~4200–5200	~4600–5100	~4500–5100	~4800–6000	~4800–6100		~3800–6500	
	Distance 19.8–25.2 km/h (m)	~20–40	~10–40	~40–120	~60–180	~80–250		~60–210	
	Distance > 25 km/h (m)	~0–2	~0–1	~0–2	~0–23	~2–30		~2–25	
[22]	Total distance (m)	-	-	-	-	-	-	-	~5611.3
	Distance 12–15 km/h (m)								~541.9
	Distance 15–20 km/h (m)								~427.8
	Distance 20–25 km/h (m)								~149.7
	Distance > 25 km/h (m)								~28.5
[24]	Total Distance (m)	-	-	-	5316.2 ± 1354.5	-	6021.4 ± 1675.6	-	4750.4 ± 1593.5
	HMLD (>25.5 W·kg^−1^) (m)				489.1 ± 228.4		730.6 ± 483.4		524.9 ± 291.4
	Distance > 25 km/h (m)				28.1 ± 41.7		130.4 ± 462.6		40.2 ± 50.4
	Distance > 25 km/h (NR)				1.9 ± 2.5		4.8 ± 4.8		3.1 ± 2.9
	Acceleration ≥ 3 m.s^−2^ (NR)				33.6 ± 18.8		53.8 ± 20.6		49.9 ± 20.2
	Deceleration ≤ −3 m.s^−2^ (NR)				30.3 ± 19.8		49.8 ± 25.1		44.0 ± 22.5
**Reference values**		**All teams**							
	**Duration (min)**	**79–117**							
	**Total distance (m)**	**3964.5–6500**							
	**Distance > 18 km/h (m)**	**11.8–250**							
	**Distance > 25 km/h (m)**	**0–30**							

U: under; min: minutes; NR: number; HMLD: high metabolic load distance.

**Table 5 healthcare-09-01567-t005:** Results for external training intensity by match-day minus and by players positions.

Study	Team and Measures	MD-5	MD-4	MD-3	MD-2	MD-1
[21]	**U12**					-
	Duration (min)	~85	~83	~88	~90	
	Total Distance (m)	~4800	~5200	~4400	~4200	
	Distance 19.8–25.2 km/h (m)	~30	~40	~30	~20	
	Distance > 25 km/h (m)	~0	~0	~2	~1	
[21]	**U13**					-
	Duration (min)	~100	~85	~85	~87	
	Total Distance (m)	~5100	~4900	~4700	~4600	
	Distance 19.8–25.2 km/h (m)	~10	~40	~20	~30	
	Distance > 25 km/h (m)	~0	~0	~0	~1	
[21]	**U14**					-
	Duration (min)	~100	~85	~83	~90	
	Total Distance (m)	~5100	~5100	~4500	~4800	
	Distance 19.8–25.2 km/h (m)	~60	~120	~40	~60	
	Distance > 25 km/h (m)	~1	~2	~0	~0	
[21]	**U15**			-		
	Duration (min)	~98	~110		~90	~85
	Total Distance (m)	~5100	~6000		~5100	~4800
	Distance 19.8–25.2 km/h (m)	~60	~180		~110	~90
	Distance > 25 km/h (m)	~0	~23		~3	~1
[21]	**U16**			-		
	Duration (min)	~93	~117		~90	~87
	Total Distance (m)	~5000	~6100		~4900	~4800
	Distance 19.8–25.2 km/h (m)	~80	~250		~120	~80
	Distance > 25 km/h (m)	~2	~30		~10	~2
[21]	**U18**			-		
	Duration (min)	~90	~115		~100	~70
	Total Distance (m)	~4900	~6500		~6000	~3800
	Distance 19.8–25.2 km/h (m)	~105	~210		~200	~60
	Distance > 25 km/h (m)	~3	~25		~23	~2
[24]	**U15–U17–U19**	-	-			
	Total Distance (m)			5372.0 ± 1452.1	5796.0 ± 1773.3	4728.0 ± 1618.6
	HMLD (m)			591.2 ± 284.9	568.2 ± 287.7	488.8 ± 293.6
	Distance > 25 km/h (m)			39.7 ± 49.1	40.4 ± 51.1	58.1 ± 76.5
	Distance > 25 km/h (NR)			3.1 ± 2.9	2.9 ± 3.7	3.8 ± 4.7
	Accelerations ≥ 3 m.s^−2^ (NR)			48.9 ± 22.8	43.6 ± 20.5	43.2 ± 19.9
	Decelerations ≤ −3 m.s^−2^ (NR)			46.0 ± 25.6	40.3 ± 20.8	34.4 ± 21.8
**Reference values**	**Total distance (m)**	**4800–5100**	**4900–6500**	**4400–5372**	**4200–6000**	**3800–4800**
	**Distance > 18 km/h (m)**	**10–105**	**40–250**	**20–40**	**20–200**	**60–90**
	**Distance > 25 km/h (m)**	**0–3**	**0–30**	**0–39.7**	**0–40**	**1–58**
**Study**	**Team and Measures**	**Central Defender**	**Wide Defender**	**Central Midfielder**	**Wide Midfielder**	**Forward**
[24]	**U15–U17–U19**					
	Total Distance (m)	5282.3 ± 1407.5	5275.9 ± 1774.6	5456.9 ± 1565.9	5370.1 ± 1692.6	5156.9 ± 1820.9
	HMLD (m)	541.3 ± 243.7	548.5 ± 282.1	602.2 ± 275.4	562.2 ± 275.4	529.5 ± 360.6
	Distance > 25 km/h (m)	44.3 ± 56.9	51.2 ± 26.6	49.1 ± 57.1	38.8 ± 48.9	56.6 ± 77.8
	Distance > 25 km/h (NR)	3.2 ± 3.3	2.7 ± 3.1	3.4 ± 3.6	3.1 ± 3.4	4.1 ± 5.1
	Accelerations ≥ 3 m.s^−2^ (NR)	44.6 ± 19.4	45.6 ± 20.0	47.8 ± 21.9	46.6 ± 24.0	45.3 ± 23.8
	Decelerations ≤ −3 m.s^−2^ (NR)	39.6 ± 18.7	40.2 ± 19.5	43.3 ± 22.2	41.2 ± 26.0	43.8 ± 34.4

U: under; min: minutes; HR: heart rate; NR: number; HMLD: high metabolic load distance; MD-: match-day minus.

**Table 6 healthcare-09-01567-t006:** Results for internal training intensity in one or two matches per week, by match-day minus and by player positions.

Study	Measures	Positions/Status/Team	Training with 1-Match Week			Training with 2 Matches-Week		
[20]			Session 1	Session 2	Session 3	Session 1	Session 2	Session 3
	RPE (CR-10, au)	Overall U15	3.9 ± 0.6	3.5 ± 0.9	3.5 ±0.6	3.4 ± 0.6	3.3 ± 0.9	2.3 ±0.5
	s-RPE (CR-10, au)	Overall U15	157.1 ± 42.2	275.7 ± 62.0	283.0 ± 47.8	313.2 ± 43.9	293.9 ± 71.1	167.2 ± 26.4
**U17**								
			**MD-5**	**MD-4**	**MD-3**	**MD-2**	**MD-1**	-
[23]	s-RPE (CR-10, au)	Wide defenders	~240	~360	~385	-	~150	-
		Central defenders	~245	~348	~390	-	~160	-
		Central midfielders	~243	~375	~400	-	~158	-
		Wide midfielders	~242	~380	~400	-	~158	-
		Forwards/strikers	~215	~355	~395	-	~152	-
		Overall team	~237	~364	~394	-	~156	-
[24]	RPE (CR-20, au)	Overall U15-U17-U19	-	-	13.3 ± 2.4	12.5 ± 1.7	13.3 ± 2.3	-
	s-RPE (CR-20, au)	Overall U15-U17-U19	-	-	1196.1 ± 211.2	1158.1 ± 221.2	1194.4 ± 205.2	-
**Study**			**Central Defender**	**Wide Defender**	**Central Midfielder**	**Wide Midfielder**	**Forward**	-
[23]	s-RPE (CR-10, au)	Overall U17	~160–390	~150–385	~158–400	~158–400	~152–395	-
[24]	RPE (CR-20, au)	Overall U15-U17-U19	44.6	44.6	47.7	46.6	45.3	-
	s-RPE (CR-20, au)	Overall U15-U17-U19	261.2	230.5	265.3	238.1	255.9	-
**Player status**								
[19]	Banister TRIMP (au)	U17 Starters	~105	-	-	-	-	-
		U17 Non-starters	~110	-	-	-	-	-
**Average training per overall team**								
[24]	RPE (CR-20, au)	Overall U15	13.7 ± 1.9	U17	15.5 ± 1.8	U19	12.5 ± 2.5	-
	s-RPE (CR-20, au)	Overall U15	1235.3 ± 171.9	U17	1215.5 ± 158.7	U19	1120.2 ± 224.7	-
[25]	RPE (CR-10, au)	U18 Gym training	5.5 ± 0.3	U18 Field Training	6.6 ± 0.6	-	-	-
	RPE (CR-10, au)	U16 Gym training	5.8 ± 0.4	U16 Field Training	6.3 ± 0.4	-	-	-
	RPE (CR-10, au)	U14 Gym training	6.3 ± 0.4	U14 Field Training	6.2 ± 0.2	-	-	-

U: under; min: minutes; MD-: match-day minus; au: arbitrary units; RPE: rated perceived exertion; s-RPE: session rated perceived exertion.

## Data Availability

Not applicable.

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
