# Peer review of "Reference Values for External and Internal Training Intensity Monitoring in Young Male Soccer Players: A Systematic Review"

_healthcare, 2021, doi:10.3390/healthcare9111567_

Round 1

Reviewer 1 Report

The submitted publication is a systematic review dealing with young male soccer players load monitoring. They performed this review by following PRISMA recommandations. Globally the study is well conducted and may be of interest for the field as it provides orders of magnitudes potentially helpful for coaches.

To help to increase the paper quality here are some comments.

Global comments: 

Maybe you could take into account the remarks of Craig et al. concerning "load" use. 

Misuse of the term ‘load’ in sport and exercise science
Craig A. et al, Journal of Science and Medicine in Sport

Specific comments 

  • introduction

This section is clear. You may explain a bit more why it is of interest to focus on young male players. For instance, why did you choose not to study publications about females?

Then as you use several acronyms in the publication you may introduce those for instance in the introduction (such as age categories U12, U13 etc). 

In the introduction you highlighted the fact that most articles were from the same team, could you add some information about this in the results and/or the discussion? Additionnally could you also highlight if there is some inter-country variability? 

  •  Methods 

This section is clear and seems to reflect a good workflow based on former recommandations. 

In the 2.3 section, could you precise what you would like to say by "were intered in various combinations" ?  

  • results

This section is mainly composed of tables. A written description of the main results would help a lot the understanding of the paper. The definition of all the acronyms would help. 

  • discussion

This section provides order of magnitudes and some elements of discussion. 

Could you please discuss the training "load" based on those kind of training? Are all kinds of trainings taken into account in those reported data? For instance, is the time training refering to the field training or also to the gym training? 

You presented differences based on age. Could you also discuss potential differences with adults ? A part of the introduction is focused on injury prevention, could you discuss the training "load" regarding potential injuries? Could you also take into account the specificity of young players 

Could you discuss why you did not take into account female players? Is there any expected differences ? 

As you mentioned that there is a well documented literature for adult training regarding player position, could you please compare / discuss what may be expected for young players? 

Could you please also discuss those results with other sports? 

I fully understand that may be difficult but could you eventually give some recommandation of the current measured load regarding what would be of interest for injury prevention and performance improvement? 

Was there any mention of players who perform several sports? And, were they taken into accoun in their "load" estimation? 

  • Conclusion

This section highlight current main findings of the paper. It may be improved when updating previous sections. 

Reviewer 2 Report

The manuscript “Reference values for external and internal load monitoring in young male soccer players: a systematic review” aimed to investigate the use of this important tool for monitoring training adaptation in youth soccer players. The methods are pristine, and the discussion is well conducted. The description of the results could be improved. The overall quality of the manuscript is good and the impact it can have for the journal and for the body of literature is significant. I commend the authors for their work. Please find specific minor comments for each section below:

The introduction is concise and clearly states the aim of the manuscript as well as the background for it.

The methods are adequate, and I commend the authors for their fine and hard work in putting together this systematic review in such a rigorous manner.

The results are a little confusing. I understand that the authors are reporting a large amount of data, but table format should be revised. Table 3 extends to another page without including headings, as an example. Please do try to revise tables for better comprehension.

In the text, the authors refer to table 3 as table 2 (first words of topic 3.3).

The discussion is concise, but clear. The obtained data is well contextualized and important insights are provided. The analyses are divided in relevant topics.

The conclusion is concise and based on the obtained data.

Round 2

Reviewer 1 Report

The article was considerably improved by the authors.